# Macrophage-Targeted Sodium Chlorite (NP001) Slows Progression of Amyotrophic Lateral Sclerosis (ALS) through Regulation of Microbial Translocation

**DOI:** 10.3390/biomedicines10112907

**Published:** 2022-11-12

**Authors:** Rongzhen Zhang, Paige M. Bracci, Ari Azhir, Bruce D. Forrest, Michael S. McGrath

**Affiliations:** 1Department of Medicine, University of California San Francisco, San Francisco, CA 94110, USA; 2Department of Epidemiology and Biostatistics, University of California San Francisco, San Francisco, CA 94158, USA; 3Neuvivo, Inc., Palo Alto, CA 94301, USA; 4Hudson Innovations, LLC, Nyack, NY 10960, USA

**Keywords:** ALS, LPS, microbial translocation, biomarker, macrophage, neuroinflammation, NP001

## Abstract

Amyotrophic lateral sclerosis (ALS) is a heterogeneous, progressive, and universally fatal neurodegenerative disease. A subset of ALS patients has measurable plasma levels of lipopolysaccharide (LPS) and C-reactive protein (CRP) consistent with low-grade microbial translocation (MT). Unless interrupted, MT sets up a self-perpetuating loop of inflammation associated with systemic macrophage activation. To test whether MT contributed to ALS progression, blood specimens from a phase 2 study of NP001 in ALS patients were evaluated for changes in activity in treated patients as compared to controls over the 6-month study. In this post hoc analysis, plasma specimens from baseline and six-month timepoints were analyzed. Compared with baseline values, biomarkers related to MT were significantly decreased (LPS, LPS binding protein (LBP), IL-18, Hepatocyte growth factor (HGF), soluble CD163 (sCD163)) in NP001-treated patients as compared to controls, whereas wound healing and immunoregulatory factors were increased (IL-10, Epidermal growth factor (EGF), neopterin) by the end of study. These biomarker results linked to the positive clinical trial outcome confirm that regulation of macrophage activation may be an effective approach for the treatment of ALS and, potentially, other neuroinflammatory diseases related to MT.

## 1. Introduction

Amyotrophic lateral sclerosis (ALS) is a universally fatal disease with motor neuron degeneration leading to paralysis and death within 2–5 years of disease onset. Approximately 10–15% of ALS patients have a familial form of the disease associated with genetic abnormalities, whereas the majority of individuals have a “sporadic” form of the disease [1,2,3]. To date, there are no approved drugs that significantly improve and extend the quality of life. 

ALS is such a heterogeneous disease that the FDA published guidelines in 2019 [4] encouraging drug developers to identify subsets of patients most responsive to their therapeutic approach. Among the many targets identified that have recently been tested clinically are those directed against ALS-associated genetic mutations [5], reactive oxygen species [6], misfolded proteins [7], dysfunctional mitochondria [8], growth factor deficiencies [9] and neuroinflammation [10]. 

Recent machine learning approaches using a genetic model of ALS (G93A SOD-1 mutant) have identified what might be the very earliest abnormalities in ALS pathogenesis [11,12,13]. Rather than the disease being initiated within the affected motor neuron body, the initial site of disease onset has been mapped to the distal axonic processes within the neuromuscular junction. The G93A ALS model has a mutant form of SOD-1; the earliest site of abnormal immune-mediated disease activity associated with this molecule is within the axonic processes and damage associated with the molecule is mediated by blood-derived macrophages [12,13]. In one study, a mutation of the macrophage “respiratory burst” gene, NADPH oxidase, led to a slowing of the disease [11]. Collectively, these data implicate peripheral blood-derived macrophages as the initiators of ALS through an attack on nerve axonal elements outside the central nervous system (CNS). Therefore, a therapeutic approach that could modify blood-derived macrophage function in ALS patients might affect disease activity. 

There has been one drug under development for the past ten years that has as its principal mechanism of action the regulation of macrophage activation though modulation of the respiratory burst function. NP001 is a proprietary formulation of sodium chlorite for intravenous use and is a regulator of innate immune function [14,15,16]. In a phase 1 dose-ascending study in ALS patients, NP001 administration was associated with a dose-dependent downregulation of the monocyte activation markers CD16 and HLA-DR [17]. Normally, phagocytes undergo an oxidative burst, producing hypochlorous acid (HOCl) when activated by infectious agents or tissue damage. This toxic byproduct is rapidly converted into taurine chloramine (TauCl), which delivers an anti-inflammatory signal to macrophages, turning off NFkB and upregulating phagocytic/wound healing functions [18,19]. NP001 chlorite, as a prodrug, is converted to TauCl when in contact with heme-associated iron, causing macrophages to become anti-inflammatory, phagocytic and wound healing [15,16]. 

A summary of NP001’s activity in two underpowered, placebo-controlled, six-month phase 2 ALS trials [10] was reported in early 2022. A post hoc analysis of the combined phase 2A [20] and phase 2B trials defined a large subset of patients who responded clinically to NP001. This subset was between 40–65 years old and had baseline plasma C-reactive protein (CRP) levels >1.13 mg/L [10]. The goal of the current pilot study was to test plasma biomarker levels in NP001-treated as compared to control patients from the phase 2A study to determine whether the clinical response to NP001 would be related to some form of macrophage-targeted immune regulation. The phase 2A study was conducted on ALS patients within 3 years of symptom onset. This study had no restrictions on plasma CRP levels, but was evaluated, in part, by assessing clinical outcomes in patients with CRP levels above and below the median value for the entire study (converted to high sensitivity CRP = 1.13 mg/L). The phase 2B study recruited participants exclusively with plasma CRP levels >1.13 mg/L. The subsequent post hoc study [10] combined results from both phase 2A and 2B in a cohort treated with 2 mg/kg NP001 with a plasma CRP value of 1.13 mg/L as a lower-level cutoff for efficacy analysis. The phase 2A study with no CRP cutoff was the only trial to investigate the role of CRP selection and plasma biomarker association on the clinical outcome. All NP001 phase 2A participants that completed the trial and had blood specimens available at baseline and six months were included in the present study. 

## 2. Materials and Methods

### 2.1. Description of ALS Phase 2A Trial and Participants

The phase 2A trial (ClinicalTrials.gov: NCT01281631, accessed on 24 January 2011) was conducted by Neuraltus Pharmaceuticals, Inc. (Palo Alto, CA, USA) from January 2011 to November 2012 at 17 sites in the United States. Details of this six-month, double-blind, placebo-controlled trial and analysis were published in 2015 [20]. Briefly, participants received a total of 20 infusions administered intravenously over 6 cycles during a 25-week, double-blind treatment period. There were 4 weeks between the start of each cycle. Cycle 1 consisted of 30 min infusions over 5 consecutive days. Cycles 2, 3, 4, 5 and 6 each consisted of 3 consecutive daily infusions. Plasma specimens for biomarker analysis were obtained at the beginning and end of the study. The age restriction defined earlier [10] was employed for the placebo and 2 mg/kg NP001 arms of the study. In total, 27 placebo (17 above CRP 1.13 mg/L, 10 below) and 28 NP001 2 mg/kg-treated (15 above CRP 1.13 mg/L, 13 below) patients qualified for the biomarker analysis. For purposes of this paper, we use the term CRP as a shortened version of high-sensitivity C-reactive protein (hs-CRP). The units expressed throughout the paper are hs-CRP units, abbreviated as CRP.

### 2.2. Analysis of Clinical Outcome Data

The primary endpoint, a change from baseline in the treated vs. placebo arms of the trial in the Revised Amyotrophic Lateral Sclerosis Functional Rating Scale (ALSFRS-R) score, of the phase 2A study was not achieved. However, when CRP plasma measurements of all participants were evaluated, the median value of the CRP level of the entire study (1.13 mg/L CRP) was used as a cutoff point for evaluating the role of inflammation in the results. The results suggested that participants with a higher level of baseline plasma CRP showed a trend towards a slowing ALSFRS-R loss over the six-month trial in patients receiving 2 mg/kg NP001 as compared to placebo. ALS patients with a plasma CRP less than 1.13 mg/L at baseline were also evaluated monthly for ALSFRS-R score loss. In measuring the loss of the ALSFRS-R score over time, a subset of patients was identified that stopped progressing over the six-month study, with the frequency in the 2 mg/kg NP001 group being higher than in the placebos by more than two-fold [20]. As presented in the combined phase 2A and 2B post hoc study, these “non-progressors” were all within a 40–65-year-old age range [10]. These results were the basis for narrowing the patient population studied in the current biomarker analysis.

### 2.3. Source of Patient Specimens

The phase 2A clinical trial plasma specimens obtained at baseline and at the end of the study were available from Neuvivo Inc. (Palo Alto, CA, USA), the biopharmaceutical company that obtained the materials from Neuraltus Pharmaceuticals, Inc. (Palo Alto, CA, USA) in 2019. The plasma specimens had been stored at −80 °C since the trial ended in 2012 and were evaluated for biomarker levels by the commercial company AssayGate, Inc. (Ijamsville, MD, USA). All patients with CRP > 1.13 mg/L within the 2 mg/kg NP001-treated and placebo arms who completed the 6-month study and had plasma available from baseline and the end of the study were included in the current biomarker study (14 participants from the 2 mg/kg NP001-treated group and 15 participants from the placebo group).

### 2.4. Plasma Factors Evaluated

The original report from phase 2A showed that plasma levels of IL-18 and lipopolysaccharide (LPS) were elevated at baseline in 2 mg/kg NP001-treated patients who showed no loss in the ALSFRS-R score over the 6-month trial as compared to treated patients who progressed during the study [18]. These plasma values in patients treated with 2 mg/kg NP001 as compared to a placebo were significantly reduced towards normal by six months. Based on these results, four groups of biomarkers were evaluated in the plasma samples at baseline and the end of the study from 14 2 mg/kg NP001-treated and 15 placebo patients: (1) microbial translocation/LPS activation-related biomarkers: LBP (LPS-binding protein), HGF (hepatocyte growth factor) and LPS; (2) classical proinflammatory factors produced by activated macrophages: IL-18, IL-6, IL-8 and TNF-a; (3) immunoregulatory and wound healing factors: IL-10, neopterin and EGF (epidermal growth factor); and (4) monocyte trafficking factor and soluble CD163 (sCD163). Baseline values for all of the biomarkers were compiled for NP001-treated patients and were compared to the placebo group for each. For outcome comparisons between treatment groups, the evaluation of the NP001 effects on plasma biomarkers was normalized to “percent of biomarker change from baseline = [100 × (biomarker value at 6 months—biomarker value at baseline)/biomarker value at baseline]”. Both the baseline values and the % change from baseline were plotted side by side for each biomarker evaluated.

### 2.5. Statistical Analyses

Statistical analysis was performed using JMP Pro 16 (SAS Institute, Cary, NC, USA). In general, data were summarized using counts and percentages for categorical data and using standard univariate descriptive statistics (number of participants, mean, standard deviation, median) for continuous data. Fisher’s exact test was used to compare percentages between placebo and NP001 with respect to non-progressors. Analysis of covariance models was used to compare the placebo to each NP001 group for continuous data. For all analyses, a two-sided *p*-value < 0.05 was considered statistically significant.

## 3. Results

### 3.1. Demographics and Clinical Characteristics of ALS Patient Groups in the Study 

The definition of ALS patients whose specimens were used to evaluate the biomarkers in the current study was established in an earlier post hoc analysis of clinical outcomes [10]. There were no significant demographic or clinical differences between ALS patients who had baseline CRP values <1.13 mg/L or >1.13 mg/L, except for the baseline plasma CRP values which were significantly different (*p* < 0.0001) (Table 1). Table 2 and Table 3 show the demographics of NP001-treated vs. placebo patients in the >1.13 CRP group and the <1.13 CRP group, respectively. Overall, there were no differences in any demographic or clinical category between any of the groups except for the baseline CRP values.

### 3.2. NP001 Significantly Slows Loss of ALSFRS-R in Treated as Compared to Placebo Patients over 6 Months

ALSFRS-R values for each patient in the study were obtained at baseline, then monthly thereafter through the 6 month trial. Baseline ALSFRS-R values for the treated and placebo arms of the CRP < 1.13 mg/L and the CRP > 1.13 mg/L groups were not statistically different from each other (Figure 1 and Figure 2, left-side box plots). In the CRP > 1.13 mg/L group, the NP001-treated arm lost 1.5 units of the ALSFRS-R score over six months, whereas the placebos lost 4.4 units (*p* = 0.03) (Figure 1). In the CRP < 1.13 mg/L group, there was no difference in the clinical outcome between the NP001-treated and placebo groups (Figure 2).

Figure 3 shows that among the 40–65-year-old and CRP > 1.13 mg/L ALS patients, those treated with 2 mg/kg NP001 had a non-progression rate of 40% as compared to the placebo patients at a less than 12% non-progression rate.

### 3.3. Plasma Biomarker Analysis 

Plasma specimens were evaluated for biomarker levels at baseline and at 6 months. All biomarker determinations were performed using the same assay system and on the same day by AssayGate. The number of patients who received 2 mg/kg NP001 or the placebo that had a baseline CRP > 1.13 mg/L, and had plasma specimens available from baseline and at 6 months for analysis, was 14 and 15, respectively. Baseline values for the NP001-treated and placebo patients were not statistically different from each other (Figure 4 and Figure 5, left column). Results for the % biomarker change from baseline were broken into three categories based on whether the biomarker values in NP001-treated vs. placebo patients: (1) decreased, (2) increased, or (3) had no change. Factors that were significantly decreased in NP001 recipients as compared to placebo over 6 months were those associated with microbial translocation (LPS, LBP and HGF), monocyte trafficking (sCD163) and inflammasome activation (IL-18) (Figure 4, right column). Biomarker values that significantly increased were those associated with either inflammation suppression (IL-10, neopterin) or wound healing (EGF) (Figure 5, right column). Plasma levels for IL-6, IL-8 and TNF-a were within the normal ranges and relatively stable over the six-month study (data not shown). 

### 3.4. Microbial Translocation Is a Pathogenic Process in Part Reversed with NP001

A schematic that describes the changes in various functions and factors that lead step by step to microbial translocation and the resolution with NP001 is shown in Figure 6. Briefly, abnormal motor neurons associated with ALS initiate a reactive inflammatory response in microglia (step 1) that causes proinflammatory factors to activate the peripheral innate immune system/macrophages to become activated. Activated macrophages are not phagocytic nor wound healing, and thus, are unable to maintain the colonic epithelial barrier integrity (step 2), allowing bacterial byproducts such as LPS to enter the blood stream (step 3) and causing systemic inflammation. Blood macrophages are attracted to the damaged CNS and migrate across the blood–brain barrier, leaving sCD163 behind in the plasma; however, as they are already activated by LPS, these cells perpetuate the MT process. The biomarker data from Figure 4 and Figure 5 show how the MT process is reversed by NP001. NP001 turns off inflammation (IL-18, IL-10 and neopterin), allowing macrophages to become wound healing, and with increased help from EGF, the colon re-establishes bowel integrity, blocking the entry of LPS into the blood (decreased HGF and LBP) and slowing migration of blood macrophages into the CNS (lower sCD163).

Microbial translocation is a self-perpetuating proinflammatory process. ALS patients show evidence for microbial translocation having detectible levels of LPS in their plasma [21]. The proposed mechanism for this process is outlined in the schematic with four steps on the left in Figure 6: (1)ALS progression advances to the point wherein abnormal neurons stimulate local microglia to produce proinflammatory factors which activate blood monocytes to become proinflammatory [22].(2)Inflammatory macrophages do not phagocytose microorganisms or participate in wound healing. The epithelial cells of the colon turn over every 5 days, making the wound healing function critical for maintaining colonic epithelium integrity. The leakage of bacteria/bacterial products occurs in the blood [23,24].(3)Bacterial endotoxins (LPS) activate blood monocytes and tissue macrophages, leading to a systemic proinflammatory state [25].(4)Monocytes are recruited from the blood into damaged tissues to initiate repair. However, LPS activation leads to trafficking of non-wound healing, non-phagocytic, proinflammatory cells into the damaged tissues, leading to persistence of the disease [22,23].

The NP001 treatment leads to the resolution of biomarker levels associated with the microbial translocation process. The proposed mechanism for this process is outlined in the right schematic of Figure 6 in four steps:(1)NP001-induced TauCl downregulates NFkB and proinflammatory factors [16,19].(2)Increased EGF augments gut epithelium repair. Bacterial translocation resolves [26,27].(3)Macrophages are converted to be phagocytic and wound healing [28].(4)Drop in sCD163 signifies that blood monocyte traffic to CNS is decreased [29].

## 4. Discussion

In 2019, the FDA published guidelines as to how results from ALS clinical trials should be analyzed [4], a document that took into account the heterogeneity of the disease and how to best evaluate clinical trial data. These FDA guidelines were applied to datasets from two ALS six-month NP001 phase 2 (phase 2A and 2B) placebo-controlled trials in a recent report [10].

Included in the original phase 2A analysis were biomarker evaluations from patients (NP001-treated vs. placebo) who showed no loss in ALSFRS-R score over the 6-month study. In the original description of the phase 2A trial, NP001 treatment led to normalization of plasma IL-18 and a reduction in plasma LPS levels in this subset [20]. The effect on LPS was, at the time, an unexpected outcome as NP001 had no ability to effect LPS levels directly. Importantly, LPS has been implicated in causing immune dysregulation, contributing to the pathogenesis of ALS and other neuroinflammatory diseases for over 10 years [21,25,30]. Based on these changes and the extensive literature on the role that chlorite conversion to taurine chloramine plays in innate immune regulation [18,19], the biomarkers evaluated in the current study were chosen as they related to macrophage activation associated with microbial translocation.

The goal of the current study was to test whether the clinical outcome would be related to biomarker changes that reflected NP001′s mechanism of action. The original phase 2A trial was underpowered to show significant clinical activity; however, patients who had plasma CRP levels above the median study value (1.13 mg/L) showed a dose-responsive relationship between NP001 and change in ALSFRS-R score over the six months [20]. In addition, in the combined phase 2A/2B analysis, the ALS patient age range of 40–65 defined patients most likely to respond to NP001 [10]. Plasma specimens evaluated in the current study were those from all participants in the phase 2A trial who received 2 mg/kg NP001 or the placebo and fell within the age range (40–65) defined in the post hoc data analysis publication [10].

The clinical outcome findings of the NP001 phase 2A study are shown in Figure 1 and Figure 2. Using the age cutoff as previously defined, clinical outcome was evaluated in ALS patients above and below the CRP 1.13 mg/L cutoff based on the change in the ALSFRS-R score over time. Note the highly significant difference between the >1.13 CRP NP001 vs. placebo patients as compared to no observable difference in patients with CRP < 1.13 mg/L. Table 1, Table 2 and Table 3 show in great detail the demographics and clinical measures associated with the patients evaluated, and except for the CRP values, everything is the same for all groups analyzed. In addition to the evaluation of the rate of ALSFRS-R score loss in the >1.13 mg/L CRP group, a 3-fold higher subset of patients stopped progressing as defined by having no loss in the ALSFRS-R score over six months (Figure 3). Based on these results, plasma specimens from the age-restricted patients with >CRP 1.13 mg/L were evaluated to test whether the clinical outcome would be related to NP001-associated biomarker changes.

Rather than test a broad spectrum of inflammatory markers, the study focused on those associated with MT [31,32]. Figure 4 shows factor levels in treated patients that at 6 months were significantly lower than in placebo patients. Changes in these factors included those regulated by NFkB, a known target of TauCl produced by the metabolism of NP001. Once LPS is removed as a perpetual driver of MT signaling, LPS reactive factors such as hepatocyte growth factor (HGF), neopterin, LPS-binding protein (LBP), IL-18 and epidermal growth factor (EGF) are up and down regulated relatively quickly [33,34,35,36]. The removal of CNS disease-associated activation signaling causes a decrease in monocyte migration into the CNS, which is reflected by decreased levels of soluble CD163 [37].

Many of the biomarkers regulated by NP001 in the study presented here are abnormal in patients with Alzheimer’s disease (AD), Parkinson’s disease (PD) and other neuroinflammatory diseases [38,39]. In a cyclic activation and self-regulation cycle that occurs with innate immune function, any measurable, unregulated factor suggests a potential use for NP001. For example, both AD [39] and PD [38] patients have lower than normal plasma EGF levels, a growth factor known to promote wound healing in the gut and restore function in animal models. 

Another factor that addresses the “chicken or the egg” phenomenon of a cyclic self-regulating pathway is neopterin. It has been long known that disease activity involving macrophage activation is associated with the upregulation of neopterin. Many studies have tried to establish the disease-promoting or suppressive effects of neopterin without clear results [35]. Through the evaluation of specimens from NP001 “responsive patients”, the neopterin changes must reflect a constructive and disease regulatory activity. Therefore, neopterin is induced by LPS but regulates the degree of innate immune activation through the induction of factors such as heme-oxygenase 1 and nrf-2 [35]. 

Potentially, the most important factor linking clinical outcome to innate immune system function is CRP. The absence of a response to NP001 below a CRP of 1.13 mg/L suggests that there are at least two pathways involved in ALS pathogenesis in relation to inflammation. The one supported by the data shown in the current study is most consistent with NP001 augmenting an innate immune system process that is already initiated in patients with slightly elevated levels of CRP. CRP is turned on in response to tissue damage and mediates an innate damage response that controls pathogenic inflammatory signals [40]. Either knocking out the CRP gene or interfering with its activity accelerates a wide variety of inflammatory diseases [41,42,43]. The CRP-mediated, anti-inflammatory process is normally driven by a response to the byproduct of oxidative burst metabolism, hypochlorous acid (HClO). The simplest interpretation of NP001′s mechanism of action is that it augments an innate immune process already being employed in patients with elevated CRP through the provision of a prodrug, chlorite, that is converted to HClO. Thus, patients with an above critical innate immune activation level of CRP are responsive to this augmentation step, whereas others are not. More detailed studies on the role innate immunity plays in ALS pathogenesis will be required to test this view of ALS pathogenesis.

In conclusion, the current study utilized a patient population clinically responsive to NP001, a macrophage activation regulator. Biomarker analysis confirmed the drug activities as regulating MT. The regulation must have occurred rapidly in these patients as a significant subset of patients halted disease progression over the 6-month study. To date, NP001 is the only drug under development that has a known mechanism of action linking biomarker changes to positive clinical outcomes in a subset of ALS patients. This study also suggests the possible role of CRP as part of an innate immune response, that when augmented with NP001 chlorite, regulates ALS pathogenesis.

## Figures and Tables

**Figure 1 biomedicines-10-02907-f001:**
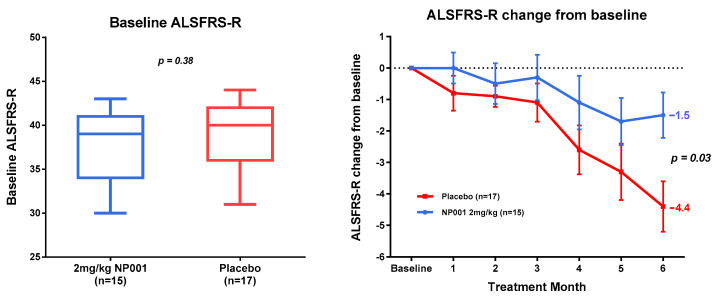
ALSFRS-R values at baseline and in response to NP001 or placebo at 6 months in those with plasma CRP > 1.13 mg/L at baseline. A comparison of baseline ALSFRS-R scores is on the left side. No statistically significant differences were found between NP001 treated and Placebo groups (Wilcoxon test, *p* = 0.38). ALSFRS-R score change from baseline is on the right side: ALSFRS-R score change from baseline for participants treated with NP001 (*n* = 15) depicted in blue and compared to the placebo group (*n* = 17) depicted in red. The plots represent the mean of ALSFRS-R score change from baseline ± SEM (standard error of the mean). Participants treated with NP001 experienced a slower decline in ALSFRS-R score from baseline compared to treatment with placebo and showed a 66% slower progression rate by the end of study (Wilcoxon test, *p* = 0.03).

**Figure 2 biomedicines-10-02907-f002:**
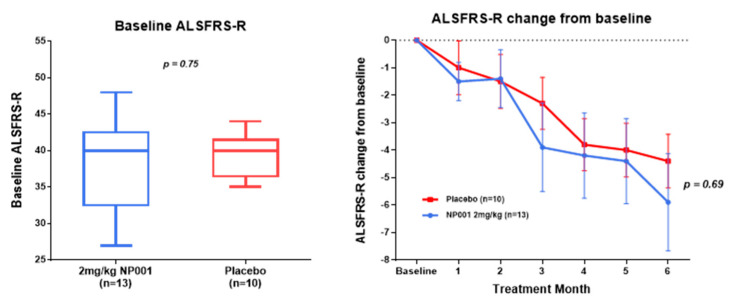
Comparison of baseline ALSFRS-R score and ALSFRS-R score change from baseline over the six-month study in participants treated with NP001 compared to placebo in those with plasma CRP < 1.13 mg/L at baseline. A comparison of baseline ALSFRS-R scores is on the left side. No statistically significant differences were found between NP001 treated and Placebo groups (Wilcoxon test, *p* = 0.75). ALSFRS-R score change from baseline is on the right side: ALSFRS-R score change from baseline for participants treated with NP001 depicted in blue (*n* = 13) and compared to placebo group depicted in red (*n* = 10). The plots represent the mean of ALSFRS-R score change from baseline ± SEM (standard error of the mean). No differences were seen be-tween NP001 and placebo groups by the end of study (Wilcoxon test, *p* = 0.69).

**Figure 3 biomedicines-10-02907-f003:**
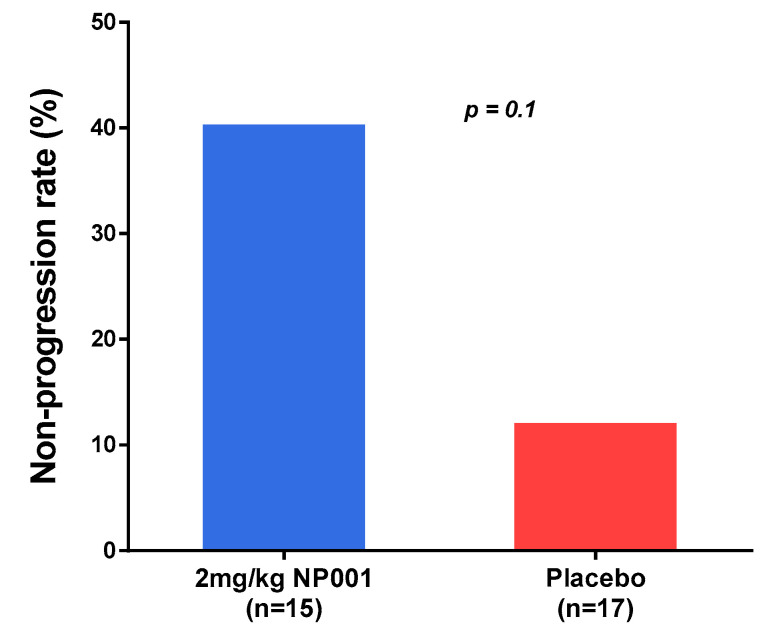
A subset of ALS patients did not progress over the 6-month study. Non-progressors were defined as those having no decrease in ALSFRS-R score at 6 months, by treatment group, restricted to those with plasma CRP > 1.13 mg/L and age between 40 and 65 years at baseline in phase 2A trial. The proportion of non-progressors (non-progression rate) in 2 mg/kg NP001 treatment (6 out of 15) was more than 3-fold higher than that of the placebo group (2 out of 17) (Fisher’s exact test, *p* = 0.1).

**Figure 4 biomedicines-10-02907-f004:**
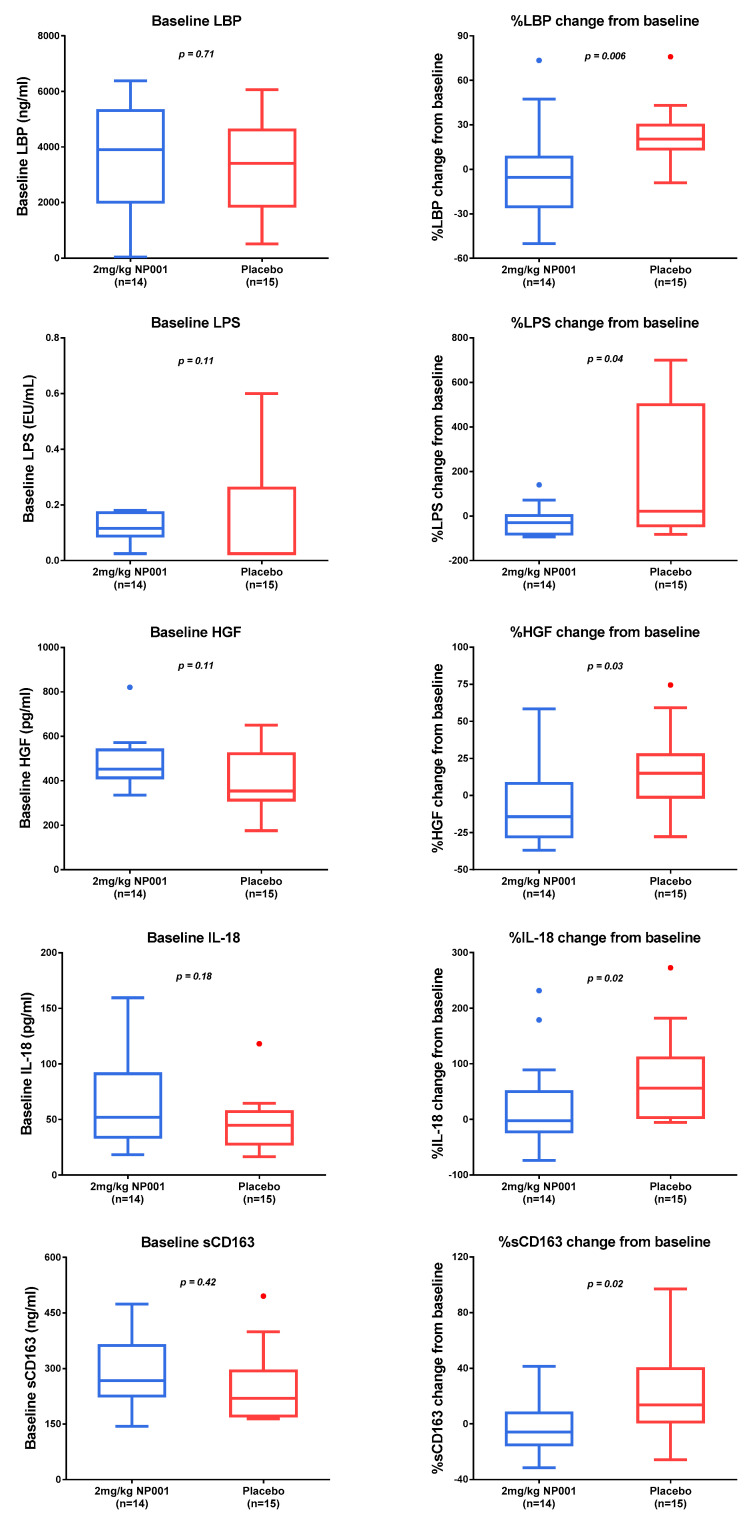
NP001-associated decrease in plasma biomarkers. Plasma biomarker levels at baseline (left column) and % change from baseline after 6 months (right column) in participants with plasma CRP > 1.13 mg/L in the phase 2A trial. Compared to the placebo group (*n* = 15), the NP001-treated (*n* = 14) group showed significantly lower levels over 6 months. The *p*-values are shown on the individual biomarker plots: (Wilcoxon test) *p* = 0.006 for LPS-binding protein (LBP), *p* = 0.04 for lipopolysaccharide (LPS), *p* = 0.03 for hepatocyte growth factor (HGF), *p* = 0.02 for IL-18 and *p* = 0.02 for soluble CD163 (sCD163).

**Figure 5 biomedicines-10-02907-f005:**
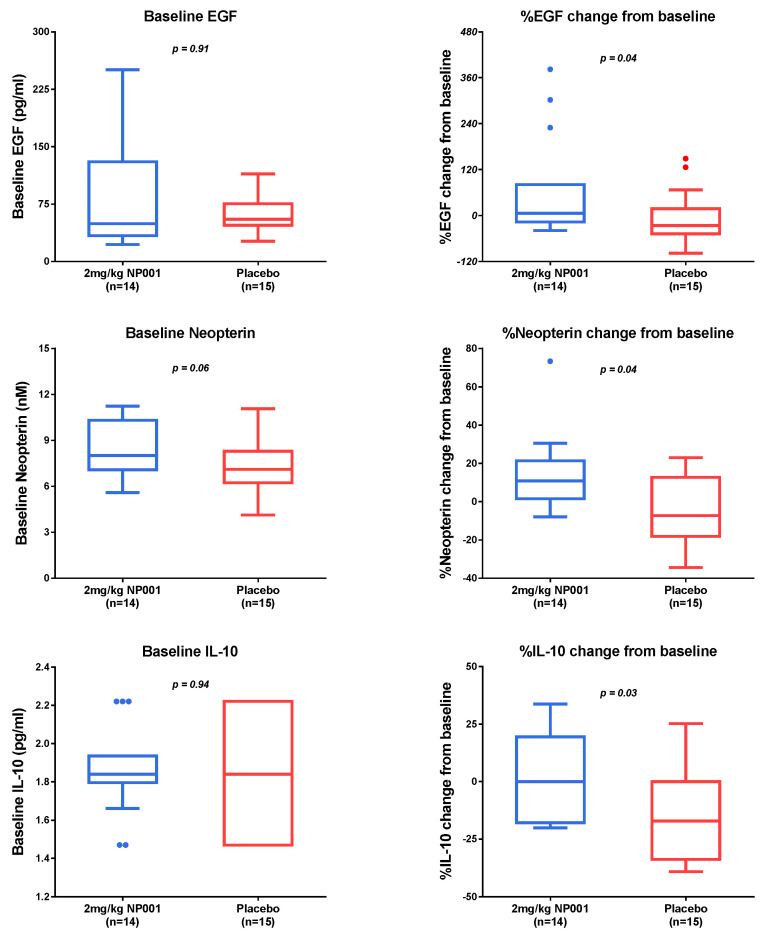
NP001-associated increase in plasma biomarkers. Plasma biomarker levels at baseline (left column) and % change from baseline after 6 months (right column) in participants with plasma CRP > 1.13 mg/L in the phase 2A trial. Compared to the placebo group (*n* = 15), the NP001-treated (*n* = 14) group showed significantly higher levels over 6 months: (Wilcoxon test) *p* = 0.03 for IL-10, *p* = 0.04 for neopterin and *p* = 0.04 for epidermal growth factor (EGF).

**Figure 6 biomedicines-10-02907-f006:**
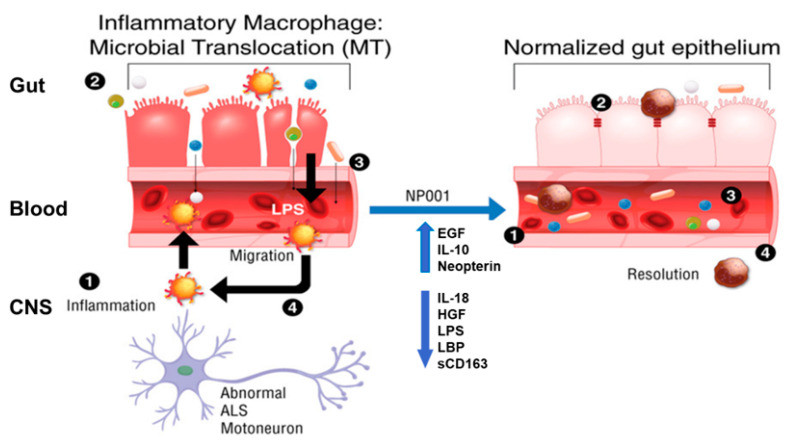
Microbial translocation and resolution with NP001. Abbreviations: CNS, central nervous system; LPS, lipopolysaccharide; EGF, epidermal growth factor; IL-10, interleukin 10; IL-18, interleukin 18; HGF, hepatocyte growth factor; LBP, LPS-binding protein; sCD163, soluble CD163. TauCl, taurine chloramine. NFkB, nuclear factor kappa B.

**Table 1 biomedicines-10-02907-t001:** Baseline demographics and characteristics of participants in groups of age between 40 and 65 years old with CRP below and above 1.13 mg/L.

	CRP > 1.13 mg/L	CRP < 1.13 mg/L	Overall
Characteristics	(*n* = 29)	(*n* = 23)	(*n* = 52)
Sex, *n* (%)			
Female	9 (31.0%)	9 (39.1%)	18 (34.6%)
Male	20 (69.0%)	14 (60.9%)	34 (65.4%)
Age at baseline, year	52.7 ± 6.7	54.7 ± 7.3	53.5 ± 7.0
Type of ALS, *n* (%)			
Familial	2 (6.9%)	3 (13.0%)	5 (9.6%)
Sporadic	27 (93.1%)	20 (87.0.0%)	47 (90.4%)
Site of ALS onset, *n* (%)			
Bulbar	6 (20.7%)	4 (17.4%)	10 (19.2%)
Limb	23 (79.3%)	19 (82.6%)	42 (80.8%)
El Escorial criteria for ALS, *n* (%)			
Definite	13 (44.8%)	11 (47.8%)	24 (46.2%)
Probable	13 (44.8%)	12 (52.2%)	25 (48.1%)
Possible	3 (10.3%)	0 (0.0%)	3 (5.8%)
Concurrent riluzole use, *n* (%)			
Yes	20 (69.0%)	15 (65.2%)	35 (67.3%)
No	9 (31.0%)	8 (34.8%)	17 (32.7%)
ALSFRS-R score at baseline ^1^, mean ± SD	38.1 ± 4.1	38.5 ± 5.1	38.3 ± 4.6
Vital capacity at baseline, mean ± SD	112.4 ± 15.6	108.8 ± 17.3	110.8 ± 16.3
Months since ALS symptom onset ^2^, mean ± SD	17.19 ± 7.80	15.73 ± 7.37	16.54 ± 7.57
CRP at baseline (mg/L) ^3^, mean ± SD	3.29 ± 2.42	0.71 ± 0.26	2.15 ± 2.22

Abbreviation: *n*, number of participants. SD, standard deviation. ^1^ ALSFRS-R score: Revised Amyotrophic Lateral Sclerosis Functional Rating Scale. ^2^ Months from ALS symptom onset to baseline. ^3^ Baseline plasma levels of C-reactive protein. There were significant differences of the actual CRP values between high- and low-CRP patient subsets (Wilcoxon test, *p* < 0.0001).

**Table 2 biomedicines-10-02907-t002:** Baseline demographics and characteristics of participants treated with NP001 vs. placebo in those with age between 40 and 65 years old and plasma CRP > 1.13 mg/L.

	NP001 2 mg/kg	Placebo	Overall
Characteristics	(*n* = 14)	(*n* = 15)	(*n* = 29)
Sex, *n* (%)			
Female	4 (28.6%)	5 (33.3%)	9 (31.0%)
Male	10 (71.4%)	10 (66.7%)	20 (69.0%)
Age at baseline, year	53.3 ± 8.1	52.1 ± 5.2	52.7 ± 6.7
Type of ALS, *n* (%)			
Familial	0 (0.0%)	2 (13.3%)	2 (6.9%)
Sporadic	14 (100.0%)	13 (86.7%)	27 (93.1%)
Site of ALS onset, *n* (%)			
Bulbar	3 (21.4%)	3 (20.0%)	6 (20.7%)
Limb	11 (78.6%)	12 (80.0%)	23 (79.3%)
El Escorial criteria for ALS, *n* (%)			
Definite	6 (42.9%)	7 (46.7%)	13 (44.8%)
Probable	7 (50.0%)	6 (40.0%)	13 (44.8%)
Possible	1 (7.1%)	2 (13.3%)	3 (10.3%)
Concurrent riluzole use, *n* (%)			
Yes	11 (78.6%)	9 (60.0%)	20 (69.0%)
No	3 (21.4%)	6 (40.0%)	9 (31.0%)
ALSFRS-R score at baseline ^1^, mean ± SD	37.3 ± 4.2	38.8 ± 4.1	38.1 ± 4.1
Vital capacity at baseline, mean ± SD	116.0 ± 15.7	109.0 ± 15.3	112.4 ± 15.6
Months since ALS symptom onset ^2^, mean ± SD	20.11 ± 7.79	14.46 ± 6.98	17.19 ± 7.80
CRP at baseline (mg/L) ^3^, mean ± SD	3.80 ± 2.60	2.82 ± 2.23	3.29 ± 2.42

Abbreviation: *n*, number of participants. SD, standard deviation. ^1^ ALSFRS-R score: Revised Amyotrophic Lateral Sclerosis Functional Rating Scale. ^2^ Months from ALS symptom onset to baseline. ^3^ Baseline plasma levels of C-reactive protein.

**Table 3 biomedicines-10-02907-t003:** Baseline demographics and characteristics of participants treated with NP001 vs. placebo in those with age between 40 and 65 years old and plasma CRP < 1.13 mg/L.

	NP001 2 mg/kg	Placebo	Overall
Characteristics	(*n* = 13)	(*n* = 10)	(*n* = 23)
Sex, *n* (%)			
Female	5 (38.5%)	4 (40.0%)	9 (39.1%)
Male	8 (61.5%)	6 (60.0%)	14 (60.9%)
Age at baseline, year	52.5 ± 7.9	57.4 ± 5.6	54.7 ± 7.3
Type of ALS, *n* (%)			
Familial	1 (7.7%)	2 (20.0%)	3 (13.0%)
Sporadic	12 (92.3%)	8 (80.0%)	20 (87.0%)
Site of ALS onset, *n* (%)			
Bulbar	2 (15.4%)	2 (20.0%)	4 (17.4%)
Limb	11 (84.6%)	8 (80.0%)	19 (82.6%)
El Escorial criteria for ALS, *n* (%)			
Definite	7 (53.8%)	4 (40.0%)	11 (47.8%)
Probable	6 (46.2%)	6 (60.0%)	12 (52.2%)
Concurrent riluzole use, *n* (%)			
Yes	9 (69.2%)	6 (60.0%)	15 (65.2%)
No	4 (30.8%)	4 (40.0%)	8 (34.8%)
ALSFRS-R score at baseline ^1^, mean ± SD	37.8 ± 6.3	39.3 ± 3.1	38.5 ± 5.1
Vital capacity at baseline, mean ± SD	110.0 ± 16.3	107.3 ± 19.4	108.8 ± 17.3
Months since ALS symptom onset ^2^, mean ± SD	14.08 ± 7.24	17.88 ± 7.34	15.73 ± 7.37
CRP at baseline (mg/L) ^3^, mean ± SD	0.70 ± 0.25	0.71 ± 0.28	0.71 ± 0.26

Abbreviation: *n*, number of participants. SD, standard deviation. ^1^ ALSFRS-R score: Revised Amyotrophic Lateral Sclerosis Functional Rating Scale. ^2^ Months from ALS symptom onset to baseline. ^3^ Baseline plasma levels of C-reactive protein.

## Data Availability

The data are available through Neuvivo, Inc. upon request.

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
