# Peer review of "Macrophage-Targeted Sodium Chlorite (NP001) Slows Progression of Amyotrophic Lateral Sclerosis (ALS) through Regulation of Microbial Translocation"

_biomedicines, 2022, doi:10.3390/biomedicines10112907_

Round 1

Reviewer 1 Report

The article by Zhang et al. demonstrates a study on the effect of sodium chlorite (NP001) treatment on plasma biomarkers of macrophage activation and microbial translocation in the context of related clinical outcomes in ALS patients. However, the study description is not well documented, the presentation of results is somewhat confusing, and the discussion requires an extension.

First of all, the authors should provide a detailed description of individual distribution in the context of the previously reported number of 136 and 138 participants in phases 2A and 2 B, respectively, with regard to NP001 dose and CRP value. Moreover, the different number of participants in Placebo groups for analysis of ALSFRS-R change from baseline in those with plasma CRP < 1.13mg/L and CRP > 1.13mg/L should also be clarified. And the absolute baseline value in both studies should be specified.

The indicated statistical significance should be labeled in Figures 1 and 2, and in Figure 3, the legend for statistical significance labeling should be provided.

It is recommended to provide relevant references in the mechanism descriptions presented in Figure 4 to support the logical flow between them. Moreover, all used symbols should be explained in a legend.

The cut-off point used in the discussion should be commented on concerning the observed different outcomes. Consequently, the conclusions should refer to this.

Author Response

Response to reviewer #1’s comments: 

  1. First of all, the authors should provide a detailed description of individual distribution in the context of the previously reported number of 136 and 138 participants in phases 2A and 2 B, respectively, with regard to NP001 dose and CRP value. Moreover, the different number of participants in Placebo groups for analysis of ALSFRS-R change from baseline in those with plasma CRP < 1.13mg/L and CRP > 1.13mg/L should also be clarified. And the absolute baseline value in both studies should be specified.

The last paragraph of the introduction and the first methods section have been revised to clarify the patient populations used in the earlier combined clinical outcome paper as compared to the pilot biomarker study presented here. Figure 1 has also been revised to include the baseline ALSFRS-R values between the treated and placebo populations for both CRP < and > 1.13.

  1. The indicated statistical significance should be labeled in Figures 1 and 2, and in Figure 3, the legend for statistical significance labeling should be provided.

The figures have been changed to incorporate the suggested changes.

  1. It is recommended to provide relevant references in the mechanism descriptions presented in Figure 4 to support the logical flow between them. Moreover, all used symbols should be explained in a legend.

Thank you for this suggestion. New references have been included to best document the steps in microbial translocation associated with neurologic diseases and steps that may be associated with resolution. Symbols as described are in the revised figure legend.

  1. The cut-off point used in the discussion should be commented on concerning the observed different outcomes. Consequently, the conclusions should refer to this.

We have included a new paragraph in the discussion that addresses the CRP cutoff point and the potential meaning in the context of the innate immune system’s response to ALS. As a marker of unknown sensitivity in the context of ALS pathogenesis, the expression of CRP in response to tissue injury is known to suppress ongoing inflammation. Of the patients evaluated in the current study, only those with a level of CRP > 1.13 mg/L appeared to both slow disease activity and change many of the factors associated with macrophage activation in the context of microbial translocation. Although not proven, innate immune reaction to evolving ALS may be both beneficial and identify a subset of patients in whom chlorite (NP001) may augment this response in a clinically beneficial manner.

Reviewer 2 Report

In the present manuscript, the data was clearly presented.

In the present study, the phase 2A trial was conducted from January 2011 through November 2012, and the plasma specimens were obtained in 2019. Had not been evaluated the biomarker levels in the past?

"EOS" in line 118 was explained in line 144. It should be explained at the first appearance.

There were many typographical errors. They should be checked and corrected in the proof. 

Author Response

Response to reviewer #2’s comments: 

  1. In the present study, the phase 2A trial was conducted from January 2011 through November 2012, and the plasma specimens were obtained in 2019. Had not been evaluated the biomarker levels in the past?

A limited number of biomarkers were evaluated in the phase 2A trial focused exclusively on the subset of patients who did not progress over the 6-month study. These data were included in the 2015 Miller et al. manuscript (Ref #20). The important observations that gave rise to the current study were the findings of detectable LPS levels that were decreased with treatment in non-progressors and that these same patients had higher baseline levels of inflammatory markers. Although blood specimens were obtained from patients in the phase 2B study, no biomarker analysis was performed.

  1. "EOS" in line 118 was explained in line 144. It should be explained at the first appearance.

EOS: “end of study” was replaced with “six months” as that was the duration of the trial.

  1. There were many typographical errors. They should be checked and corrected in the proof.

Typos were corrected.

Reviewer 3 Report

In this manuscript, the authors highlighted significant results by a post hoc analysis of the combined phase 2A and phase 2B trials probing the effectiveness of NP001 chlorite, as a disease-modifying drug in patients with amyotrophic lateral sclerosis (ALS). More in detail, in a selected group of ALS patients, the authors assessed the relationships between the molecular activity of NP001, as shown by the reduction of IL-18 and LPS plasma levels (i.e., anti-inflammatory signal to macrophages), and the clinical effect of disease slowness, as demonstrated by the ALSFRS scores. Results provided by the biomarker analysis confirmed the significant effect of NP001 in modifying innate inflammation.

The study is interesting and well-designed. It is based on the strong rationale of an inflammatory disorder as the pathophysiologic underpinning of ALS as also pointed out by genetic results. Overall, the NP001 seems a promising disease-modifying drug that must be validated by following studies. However, I have several concerns about the study.

First, the authors should point out that the rationale beyond the use of such drugs would be applicable only to a subgroup of well-selected patients, probably those in which the role of disrupted innate inflammation is well documented for instance by genetic testing.

The authors should provide details about pros and cons of detecting immunologic compounds relative to macrophagic function

in blood samples.

It is unclear whether the clinical phenotype of ALS would be relevant for determining the NP001 response.

The authors should provide further details on the clinical assessment of patients.

More details about pharmacologic schedule should be provided.

It is unclear whether patients were taking any further drug acting on the CNS.

Author Response

Response to reviewer #3’s comments: 

  1. First, the authors should point out that the rationale beyond the use of such drugs would be applicable only to a subgroup of well-selected patients, probably those in which the role of disrupted innate inflammation is well documented for instance by genetic testing.

We’ve provided commentary as to the target patient profile based on the subset that had their disease slowed or stopped during the trial in the discussion. In general, these patients have > 1.13 mg/L CRP and have had ALS symptoms for more than 1 ½ years when treated. Given that CRP is an active component of the innate immune system and NP001 response tracks with CRP, we’ve added a paragraph to the discussion regarding the role that the innate immune system and potential dysfunction plays in ALS.

  1. The authors should provide details about pros and cons of detecting immunologic compounds relative to macrophagic function in blood samples.

We’ve added a paragraph to the discussion that speaks to the balance between macrophage activation and differentiation state as it occurs normally and in the context of disease.

  1. It is unclear whether the clinical phenotype of ALS would be relevant for determining the NP001 response.

In this study and in the combined post hoc study (10) of patients in the phase 2A and 2B there doesn’t seem to be a clinical phenotype that tracks in a predictable manner with a NP001 response. The different categories of responses include: 1) the most restrictive are the patients who do not progress in ALSFRS-R score loss over the 6 months of study. That percent is 35-40% with placebos being in the 10% range. 2) The most generalized of the responses are the % change in VC where most of the patients have a slowing of respiratory function loss (10). Other than the age limitation of < = 65 and the CRP > 1.13 mg/L, no obvious clinical characteristics define a responder phenotype.

  1. The authors should provide further details on the clinical assessment of patients.

We have added a table that describes the demographics and clinical characteristics of the patient groups studied.

  1. More details about pharmacologic schedule should be provided.

The protocol for NP001 administration is described in the methods section 2.1. The proportion of patients taking riluzole is in the new demographics table.

  1. It is unclear whether patients were taking any further drug acting on the CNS.

This study was performed in 2011 and the only ALS drug available was riluzole. Patients with other medical indications requiring CNS medications were excluded from the trial as per protocol (Miller 2015, Ref #20).

Round 2

Reviewer 1 Report

There is still a lack of relevant labeling significant changes in figures, like for the comment that „NP001 significantly slows loss of ALSFRS-R in treated as compared to placebos over 6 months”

Moreover, there is a need to specify what data represent in figures (average value?). Please note that in Figure 2, no SD or SEM bars have been provided.